# SNPs Sets in Codifying Genes for Xenobiotics-Processing Enzymes Are Associated with COPD Secondary to Biomass-Burning Smoke

Enrique Ambrocio-Ortiz [1,2] , Gloria Pérez-Rubio [1] , Alejandra Ramírez-Venegas [3],
Rafael de Jesús Hernández-Zenteno [3], Juan Carlos Fernández-López [4] , María Elena Ramírez-Díaz [5],
Filiberto Cruz-Vicente [6], María de Lourdes Martínez-Gómez [7], Raúl Sansores [8], Julia Pérez-Ramos [9,*]
and Ramcés Falfán-Valencia [1,*]

1 HLA Laboratory, Instituto Nacional de Enfermedades Respiratorias Ismael Cosío Villegas,
  Mexico City 14080, Mexico
2 Doctorado en Ciencias Biológicas y de la Salud, Universidad Autónoma Metropolitana-Xochimilco,
  Calzada del Hueso 1100, Col. Villa Quietud, Coyoacán, Ciudad de México 04960, Mexico
3 Tobacco Smoking and COPD Research Department, Instituto Nacional de Enfermedades Respiratorias Ismael
  Cosio Villegas, Mexico City 14080, Mexico
4 Computational Genomics Department, Instituto Nacional de Medicina Genómica, Mexico City 14610, Mexico
5 Coordinación de Vigilancia Epidemiológica, Jurisdicción 06 Sierra, Tlacolula de Matamoros Oaxaca,
  Servicios de Salud de Oaxaca, Oaxaca 70400, Mexico
6 Internal Medicine Department, Hospital Civil Aurelio Valdivieso, Servicios de Salud de Oaxaca,
  Oaxaca 68050, Mexico
7 Hospital Regional de Alta Especialidad de Oaxaca, Oaxaca 71256, Mexico
8 Clínica de Enfermedades Respiratorias, Fundación Médica Sur, Mexico City 14080, Mexico
9 Departamento de Sistemas Biológicos, Universidad Autónoma Metropolitana-Xochimilco,
  Calzada del Hueso 1100, Col. Villa Quietud, Coyoacán, Ciudad de México 04960, Mexico
* Correspondence: jperez@correo.xoc.uam.mx (J.P.-R.); rfalfanv@iner.gob.mx (R.F.-V.)

**Abstract:** Chronic obstructive pulmonary disease (COPD) is the third leading cause of death world-wide; the main risk factors associated with the suffering are tobacco smoking (TS) and chronic exposure to biomass-burning smoke (BBS). Different biological pathways have been associated with COPD, especially xenobiotic or drug metabolism enzymes. This research aims to identify single nucleotide polymorphisms (SNPs) profiles associated with COPD from two expositional sources: tobacco smoking and BBS. One thousand-five hundred Mexican mestizo subjects were included in the study and divided into those exposed to biomass-burning smoke and smokers. Genome-wide exome genotyping was carried out using Infinium Exome-24 kit arrays *v.* 1.2. Data quality control was conducted using PLINK 1.07. For clinical and demographic data analysis, Rstudio was used. Eight SNPs were found associated with COPD secondary to TS and seven SNPs were conserved when data were analyzed by genotype. When haplotype analyses were carried out, five blocks were predicted. In COPD secondary to BBS, 24 SNPs in *MGST3* and CYP family genes were associated. Seven blocks of haplotypes were associated with COPD-BBS. SNPs in the *ARNT2* and *CYP46A1* genes are associated with COPD secondary to TS, while in the BBS comparison, SNPs in *CYP2C8*, *CYP2C9*, *MGST3*, and *MGST1* genes were associated with increased COPD risk.

**Keywords:** chronic obstructive pulmonary disease; toxicity; indoor pollution; microarray analysis; genome-wide association study

## 1. Introduction

Chronic obstructive pulmonary disease (COPD) is a complex and multifactorial disease; preventable, treatable, and partially reversible, characterized by airflow limitation due to an airway inflammatory process in response to chronic exposure to noxious particles [1–3]. Worldwide, COPD is the third leading cause of death, with a prevalence of

251 million people and nearly 3000 million deaths. Of these deaths, 90% were recorded in low- and middle-income countries [3,4].

COPD is associated with environmental exposure risk factors, such as tobacco smoking or biomass-burning smoke exposure (BBS). Other factors are clinical and genetic characteristics that could lead to different phenotypes of COPD [1,2].

The main clinical features of COPD are dyspnea, pulmonary hypertension, hypoxemia, hypercapnia, chest tightness, wheezing, cough, phlegm (sputum), mucus (bronchitis), and low oxygen saturation, among others. [2,5] It has been described that the clinical phenotype can vary depending on the exposure factor. For example, patients with COPD secondary to BBS (COPD-BBS) present higher FEV1 values than smokers with COPD (COPD-S) [6,7]; the predominant phenotype in COPD-BBS is bronchitis, with increased production of mucus and phlegm [8]. In addition, the CAT (COPD assessment test) score indicates that patients with COPD-BBS have a smaller decrease in their quality of life, evidenced by less difficulty when carrying out their daily activities and better physical performance (6-min walk test) [9]. These phenotypic differences in the disease may be due to the heterogeneity of pollutants substances' content and individuals' genetic variability.

COPD research has dabbled in the genetics of the disease. So far, it is known that the genetic deficiency of α1-antitrypsin, encoded by the *SERPINA1* gene, caused by a single nucleotide polymorphism (SNPs), is the only genetic factor that predisposes to COPD, regardless of environmental risk factors [10]; however, it does not explain the whole variety of the disease in the world population.

Genome-wide association studies (GWAS), in which thousands of genetic variants are included in case-control comparisons and cohort models, have helped identify SNPs' association with the disease's clinical characteristics, such as tobacco smoking, cachexia, and decreased lung function [11–13]. Some associated SNPs are present in non-coding areas, making it challenging to explain their biological impact on the disease.

Another strategy used in genetic association studies is the analysis of variants in candidate genes by gene functionality or their relationship with other genes within the same interaction network [14,15]. Some reported genes include *TNF*, *CCL2*, *SERPINA12*, *SERPINE2*, *FAM13A*, and *TFGB1*; this has allowed the description of associations between SNPs and the different clinical phenotypes of the disease (emphysema or bronchitis) [16,17]. It is essential to mention that these studies have been carried out in groups of smokers, which has left aside the evaluation of patients with COPD-BBS.

This work aims to describe the association of SNPs in candidate genes related to the processing of xenobiotic, cytotoxic, and drugs with COPD, both secondary to tobacco smoking and BBS, and possible SNPs profiles that can differentiate them.

## 2. Materials and Methods

### 2.1. Population Included

For this study, 1500 subjects were included and divided into 2 comparison groups; the first was composed of 900 smokers: 300 patients with COPD secondary to tobacco smoking (COPD-S) and 600 smokers without the disease (SWOC). The second comparison group included 600 subjects exposed to smoke from biomass burning, divided into 220 patients with COPD secondary to BBS (COPD-BBS) and 380 subjects exposed to BBS but without COPD (BBES).

Mexican mestizo subjects over 40 years of age and of indistinct sex were included; for the comparison group of smokers, participants with a tobacco index (TI) > 5 packs/year and no history of exposure to BBS were included. In the exposed to BBS group, subjects with an exposure index to BBS (BEI) >100 h/year were included and were never smokers. COPD patients were defined when the post-bronchodilator FEV1/FVC ratio was <70% (Supplementary Figure S1). Participants with other inflammatory, autoimmune, or respiratory diseases were eliminated. The participants were recruited from the COPD Clinic, smoking cessation support groups of the Tobacco Smoking and COPD Research Department, from COPD early detection campaigns in rural Oaxaca [18], and from suburban areas

of Mexico City. The Ethics in Research Committee from Instituto Nacional de Enfermedades Respiratorias Ismael Cosio Villegas approved the protocol under code numbers B14-17, B11-19, and C53-19.

The clinical evaluation of the patients was carried out by specialized chest physicians from the Instituto Nacional de Enfermedades Respiratorias Ismael Cosio Villegas using GOLD guidelines. Ref. [19] Demographic and ancestry data were obtained through a questionnaire. Before taking biological samples, all patients signed an informed consent approved by institutional ethical boards.

### 2.2. Biological Samples

All participants took a blood sample by forearm puncture, and DNA and plasma were obtained using the previously described methodology [20]. The DNA samples were quantified spectrophotometry through a nanodrop device (Thermo Scientific, Wilmington, DE, USA), and samples with a 260/280 ratio between 1.8 and 2.2 were selected, adjusted to 60 ng/µL, and their integrity was evaluated in 1.5% agarose gels.

### 2.3. Whole Exome Genotyping

After sample quality control, the groups were composed of 370 smokers (COPD-S = 150, SWOC = 220) and 401 subjects exposed to BBS (COPD-BBS = 101, BBES = 300) (Supplementary Figure S2). The samples were genotyped using the Illumina Infinium Exome-24 Kit arrays v1.2 (5200 Illumina Way, San Diego, CA 92122, USA) with a genotyping capacity of up to 560,000 variants.

We applied functional candidate gene methodology to select only genes related to the metabolism of xenobiotics, cytotoxic products, and drug metabolism. Through bibliography research, we selected 38 genes. After applying the Hardy–Weinberg ($p > 1 \times 10^{-9}$ test and excluding SNPs with MAF < 10%, we chose all SNPs in the proposed genes. We worked with 748 SNPs in both comparison groups (Supplementary Figure S3).

### 2.4. Data Analysis

PLINK v. 1.07 [21] was used for data quality control (QC). We considered a genotype call rate > 95% and eliminated subjects with >0.05 of missing genotypes; sex discrepancies were considered by X chromosome homozygosity (men > 0.8, women < 0.2), while relatedness was assessed by identity by descent (IBD) considering pi-hat values < 0.25.

Association analysis was carried out using PLINK v. 1.07 applying Fisher's exact test adjusted by covariates; in the smoker group comparison, we included sex, age, and cigarettes/day (TI) as covariates, while in the biomass group, comparison, age, and biomass burning-smoke exposure index (BEI) was included, utilizing the Bonferroni correction test.

The R language [22] and the Rstudio interface [23] were employed for statistical analysis. Admixture and principal component analysis (PCA) were carried out using packages SNPRelate and gdsfmt from Bioconductor. We included Hapmap population data from Northern Europeans from Utah (CEU), Yoruba in Ibadan from Nigeria (YRI), and native Amerindian populations (AMR) described by Huerta-Chagoya, and we selected 32 ancestry informative markers (AIMs) and used k = 3 [24]. The distribution of demographic variables, exposure data, or lung function was analyzed to determine the type of statistical comparison being made.

### 2.5. Severity Analysis

Afterward, we stratified COPD patients based on the GOLD states, comparing mild (GOLD 1 + 2) vs. severe forms (GOLD 3 + 4) of the illness to avoid bias by subgrouping. This analysis was carried out for COPD-S and COPD-BBS individually using PLINK v. 1.07 and applying Fisher's exact test, correcting by covariates, and the Bonferroni multiple testing.

### 2.6. Multiple Correspondence Analysis

We applied multiple correspondence analysis (MCA) to determine possible grouping between SNPs associated, exposure indexes, and/or FEV1 values. These analyses were carried out using Rstudio [22] and the packages FactoMineR [25] and factoextra [26].

### 2.7. Calculation of Haplotype Blocks

We included 750 SNPs in the Haplotype blocks analysis. This analysis was carried out using Haploview 4.2 software, [27] applying the analysis algorithm presented by Gabriel et al. [28]. We applied a window of inclusion of 5000 Kb per pair of SNPs. Linkage disequilibrium (LD) was presented using D' value. Haplotypes association analysis was carried out using R through Fisher's exact test between cases vs. controls and adjustment by logistic regression, including covariates. Genes' schemes and SNPs' positions are included in the Supplementary Material (Supplementary Figures S7 and S8).

## 3. Results

### 3.1. Population Studied

After quality control, 745 subjects were included; 354 were in the group of smokers (COPD-S = 141, SWOC = 213) and 391 had been exposed to BBS (COPD-BBS = 98, BBES = 293). The distribution of the variables presented a non-normal distribution, so the demographic, clinical, and exposure variables are presented as a function of the median and quartiles 1 and 3. At the same time, the comparisons were made using the Mann–Whitney U test and $\chi^2$ for qualitative variables.

When comparing COPD-S vs. SWOC, significant differences ($p < 0.05$) were found in the male–female ratio, age, BMI, and TI; because of this, sex, age, and TI were selected for covariate correction. In the BBS comparison group (COPD-BBS vs. BBES), significant differences ($p < 0.05$) were found in age and BBS exposure index (BEI), so these were included as covariates in the association analysis of this group (Table 1).

**Table 1.** Demographic and lung function data of COPD and control subjects.

| | COPD-S (n = 141) | SOWC (n = 213) | *p* | COPD-BBS (n = 98) | BBES (n = 293) | *p* |
|---|---|---|---|---|---|---|
| *Demographic data* | | | | | | |
| Sex (M/W)% | 73.7/26.3 | 51.8/48.2 | <0.01 † | 90/10 | 99.3/0.7 | 0.57 † |
| Age (Years) | 68 (62–74) | 51 (44–58) | <0.01 | 73 (68–78) | 61 (54–69) | <0.01 |
| BMI (Kg/m$^2$) | 25.5 (22.7–29.3) | 27.6 (25.0–30.1) | 0.04 | 26.1 (23.0–31.2) | 27.6 (24.7–30.8) | 0.09 |
| *Tobacco smoking data* | | | | | | |
| Cigarette per day (cig/day) | 20 (12–30) | 16 (10–21) | 0.64 | | | |
| Years of smoking (Years) | 41 (32.0–50.0) | 30 (24.0–37.5) | <0.01 | | | |
| TI (pack/year) | 40 (21.0–54.5) | 25 (16.5–39.0) | <0.01 | | | |
| *Biomass-burning smoke exposure data* | | | | | | |
| Hours or exposition (h/day) | | | | 12 (10.0–15.0) | 10 (10.0–12.0) | <0.01 |
| Years of exposition (years) | | | | 50 (33.5–60.0) | 40 (15.0–53.0) | <0.01 |
| BBS smoke exposition index (BEI) | | | | 453.0 (350.0–600.0) | 400.0 (150.0–530.0) | <0.01 |
| *Lung function data (post-bronchodilator)* | | | | | | |
| FEV$_1$ (%) | 58.0 (43.0–76.0) | 96.5 (86.0–106.0) | <0.01 | 68.0 (54.0–81.0) | 103.0 (93.0–115.0) | <0.01 |
| FVC (%) | 83.0 (71.0–98.0) | 91.5 (86.0–104.0) | <0.01 | 87.0 (74.0–100.0) | 99.0 (87.0–110.5) | <0.01 |
| FEV$_1$/FVC (%) | 57.6 (44.9–64.9) | 81.5 (78.0–85.4) | <0.01 | 60.7 (50.9–67.0) | 84.5 (78.0–93.6) | <0.01 |
| *GOLD state%* | | | | | | |
| GOLD I (%) | 15 (11.1) | | | 28 (32.2) | | NA |
| GOLD II (%) | 73 (54.1) | | | 47 (54.0) | | NA |
| GOLD III (%) | 33 (24.4) | | | 11 (12.6) | | NA |
| GOLD IV (%) | 14 (10.4) | | | 1 (1.2) | | NA |

Demographic, tobacco consumption, biomass-burning smoke exposition, and lung function data are expressed in median and quartiles (q1–q3). Qualitative variables were compared by $\chi^2$ test (†), and qualitative variables by the Mann–Whitney U test. Statistical differences were considered when $p < 0.05$. Kg: kilograms; m: meters; TI: tobacco index; FEV$_1$: forced expired volume in 1st second; FVC: forced vital capacity; NA = not applicable.

By ancestry analysis, we found different proportions for both groups of comparison. We found a highly conserved Amerindian composition in the biomass-burning comparison

group (COPD_BBS, BBES), while in the smokers' comparison (COPD_S, SWOC), we found a heterogeneous composition, predominantly Amerindian and Caucasian (Figure 1).

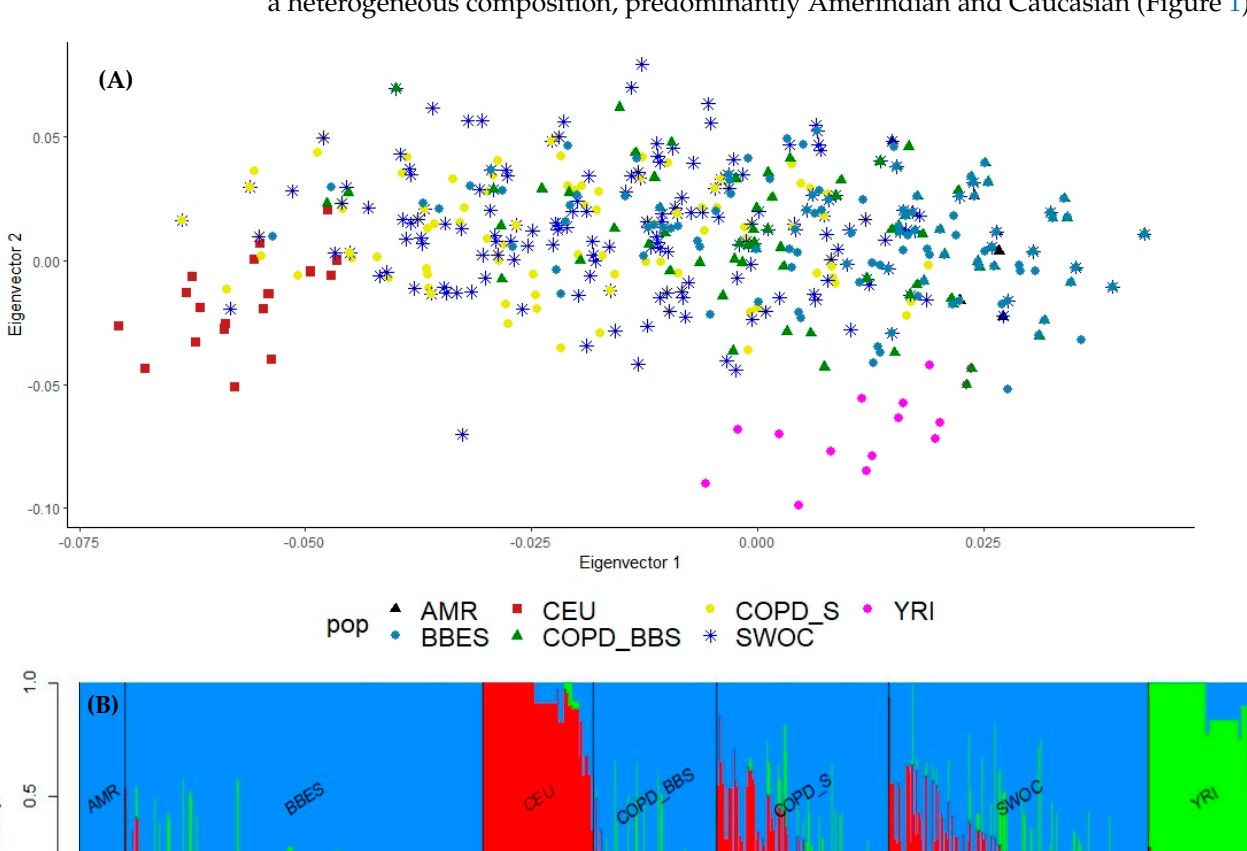

**Figure 1.** (**A**) PCA for the ancestral composition of the included population. We included Hapmap population data: northern Europeans from Utah (CEU), Yoruba in Ibadan from Nigeria (YRI), and native Amerindian (AMR). COPD_BBS: COPD patients exposed to biomass-burning smoke; COPD_S: COPD patients as smokers; SWOC: Smokers without COPD; COPD_BBS: COPD secondary to biomass-burning smoke exposition; BBES: Biomass burning exposed subject. (**B**) Admixture plot for ancestry composition in subjects included. We included the following Hapmap population data: Northern Europeans from Utah (CEU, in red), Yoruba in Ibadan from Nigeria (YRI, in green), and native Amerindian (AMR, in blue).

### 3.2. Association Analysis in the Group of Smokers

All the SNPs associated with this stage met the Hardy–Weinberg equilibrium and MAF > 10% (Supplementary Table S1). In the comparison of smokers (COPD-S vs. SWOC), after correction for covariates, 8 SNPs/alleles were found associated ($p < 0.05$) with COPD secondary to TS, 6 SNPs (rs11572191, rs8133, rs17497857, rs4964059, rs3901896, rs8041826) associated with increased risk (OR > 1.0), and 2 SNPs (rs4147611, rs3742377) with decreased risk (OR < 1.0). Of the SNPs associated with risk, rs11572191 in the *CYP2J2* gene presented the highest OR value, with an almost three-fold increased risk of developing COPD secondary to smoking. On the other hand, the *ARNT2* and *ARNTL2* genes each presented two SNPs associated with increased risk, these being the genes with the highest number of associated SNPs in this comparison group. However, when we applied the Bonferroni correction, no significant associations were retained (Table 2).

Seven SNPs associated with an increased risk of COPD secondary to TS were found in the genotype analysis. Of these SNPs, rs11572191 in *CYP2J2*, rs17497857 in *ARNTL2*,

rs3901896, and rs8041826 in *ARNT2* remained associated. On the other hand, the rs1951576 and rs943881 in *CYP46A1* and rs6488842 in *MGST1* are new findings by this model analysis. Interestingly, rs11572191 and rs17497857 are associated with heterozygous genotypes (Table 3).

**Table 2.** Allele association analysis in smokers' comparison.

| SNP/Alleles | COPD-S (n = 141) | AF% | SWOC (n = 213) | AF% | OR | CI (95%) | *p* | *p* * |
|---|---|---|---|---|---|---|---|---|
| | | | rs11572191/*CYP2J2* | | | | | |
| C | 245 | 86.88 | 401 | 94.13 | | 1.00 (Ref.) | | |
| T | 37 | 13.12 | 25 | 5.87 | 2.96 | 1.67–5.26 | 0.0002 | NS |
| | | | rs8133/*MGST3* | | | | | |
| G | 197 | 69.86 | 335 | 78.64 | | 1.00 (Ref.) | | |
| T | 85 | 30.14 | 91 | 21.36 | 1.48 | 1.05–2.08 | 0.026 | NS |
| | | | rs4147611/*MGST3* | | | | | |
| G | 163 | 57.80 | 205 | 48.12 | | 1.00 (Ref.) | | |
| T | 119 | 42.20 | 221 | 51.88 | 0.71 | 0.52–0.96 | 0.026 | NS |
| | | | rs17497857/*ARNTL2* | | | | | |
| T | 246 | 87.23 | 392 | 92.02 | | 1.00 (Ref.) | | |
| A | 36 | 12.77 | 34 | 7.98 | 1.75 | 1.03–2.96 | 0.037 | NS |
| | | | rs4964059/*ARNTL2* | | | | | |
| A | 217 | 76.95 | 352 | 82.63 | | 1.00 (Ref.) | | |
| C | 65 | 23.05 | 74 | 17.37 | 1.50 | 1.01–2.24 | 0.047 | NS |
| | | | rs3742377/*CYP46A1* | | | | | |
| G | 240 | 85.11 | 336 | 78.87 | | 1.00 (Ref.) | | |
| A | 42 | 14.89 | 90 | 21.13 | 0.61 | 0.39–0.92 | 0.019 | NS |
| | | | rs3901896/*ARNT2* | | | | | |
| T | 137 | 48.58 | 243 | 57.04 | | 1.00 (Ref.) | | |
| C | 145 | 51.42 | 183 | 42.96 | 1.40 | 1.03–1.90 | 0.029 | NS |
| | | | rs8041826/*ARNT2* | | | | | |
| A | 238 | 84.40 | 378 | 88.73 | | 1.00 (Ref.) | | |
| G | 44 | 15.60 | 48 | 11.27 | 1.61 | 1.03–2.53 | 0.037 | NS |

Comparison of frequencies by logistic regression method, including age, sex, and TI as covariates. We considered a significant association when *p* < 0.05. * *p*-value after Bonferroni correction test, AF: allele frequency, %. OR: odds ratio; CI: confidence interval; SNP: single nucleotide polymorphism; Ref: reference; NS: no significant.

We extracted the data of COPD-S and COPD-BBS, looking for possible differentiation patterns, including SNPs with MAF > 1% by MCA. Even though we have differential grouping patterns, the variance did not surpass >1% (Supplementary Figure S4A).

We included 336 SNPs for the MCA in smokers' comparison. By biplots, we did not find any cluster of SNPs that could explain variance >1% (Supplementary Figure S4C). Next, we included all the SNPs associated with the allele analysis but did not get any possible component (Supplementary Figure S5).

The possible participation of other SNPs in the genetic susceptibility was assessed through haplotype blocks, including all associated SNPs, before correction for covariates to maximize the analysis screen. Five blocks of haplotypes were found to form in the *ARNTL2* gene, *CYP19A1*, *ARNT2*, *CYP46A2*, and *MGST3*, all with LD > 85 (Figure 2).

When the association of haplotypes was carried out, we found nine different combinations of SNPs associated with COPD-S in the genes: *ARNTL* (rs10741616-rs7126796), *ARNTL2* (rs11048977-rs1037924-rs17497857-rs7138982), *CYP19A1* (rs10046-rs700519-rs6493489-rs2899472-rs2414095-rs700518), *ARNT2* (rs1374213-rs3901896-rs7168908-rs2278709), *CYP46A1* (rs3742377-rs943881-rs1951576-rs12435918-rs2146238), *ARNT* (rs10847-rs11552229-rs2228099), and *MGST3* (rs8133-rs4147611) (Table 4). Of these combinations, six haplotypes were associated with a lower risk of COPD-S and three to higher risk (OR > 1.5). We found five haplotypes containing SNPs previously associated in the allele or genotype analysis: rs3901896 in *ARNT2*, rs1951576 in *CYP46A1*, and rs17497857 in *ARNTL2*, also rs8133 and rs4147611 in *MGST3*.

**Table 3.** Genotype association analysis in smokers' comparison.

| SNP/Alleles | COPD-S (n = 141) | GF% | SWOC (n = 213) | GF% | OR | CI (95%) | *p* |
|---|---|---|---|---|---|---|---|
| | | | rs11572191/*CYP2J2* | | | | |
| CC | 105 | 74.47 | 190 | 89.20 | | 1.00 (Ref.) | |
| CT | 35 | 24.82 | 21 | 9.86 | 5.51 | 2.36–13.5 | 0.0001 |
| TT | 1 | 0.71 | 2 | 0.94 | 2.5 | 0.007–0.08 | 0.86 |
| | | | rs17497857/*ARNTL2* | | | | |
| TT | 106 | 75.18 | 182 | 85.45 | | 1.00 (Ref.) | |
| TA | 34 | 24.11 | 28 | 13.15 | 2.41 | 1.14–5.19 | 0.022 |
| AA | 1 | 0.71 | 3 | 1.41 | 1.17 | 1.89–39.23 | 0.94 |
| | | | rs3901896/*ARNT2* | | | | |
| TT | 38 | 26.95 | 70 | 32.86 | | 1.00 (Ref.) | |
| TC | 61 | 43.26 | 103 | 48.36 | 1.05 | 0.51–2.15 | 0.89 |
| CC | 42 | 29.79 | 40 | 18.78 | 2.75 | 1.19–6.56 | 0.019 |
| | | | rs8041826/*ARNT2* | | | | |
| AA | 103 | 73.05 | 168 | 78.87 | | 1.00 (Ref.) | |
| AG | 32 | 22.70 | 42 | 19.72 | 2.49 | 1.17–5.39 | 0.019 |
| GG | 6 | 4.26 | 3 | 1.41 | 4.0 | 5.98–41.7 | 0.0002 |
| | | | rs1951576/*CYP46A1* | | | | |
| AA | 85 | 60.28 | 133 | 62.44 | | 1.00 (Ref.) | |
| AG | 41 | 29.08 | 73 | 34.27 | 0.73 | 0.37–1.40 | 0.35 |
| GG | 15 | 10.64 | 7 | 3.29 | 3.88 | 1.02–15.16 | 0.047 |
| | | | rs6488842/*MGST1* | | | | |
| CC | 79 | 56.03 | 122 | 57.28 | | 1.00 (Ref.) | |
| CT | 46 | 32.62 | 82 | 38.50 | 1.03 | 0.55–1.95 | 0.92 |
| TT | 16 | 11.35 | 9 | 4.23 | 3.85 | 1.16–13.21 | 0.029 |
| | | | rs943881/*CYP46A1* | | | | |
| TT | 79 | 56.03 | 122 | 57.28 | | 1.00 (Ref.) | |
| TC | 46 | 32.62 | 82 | 38.50 | 1.03 | 0.55–1.95 | 0.92 |
| CC | 16 | 11.35 | 9 | 4.23 | 3.85 | 1.16–13.21 | 0.029 |

Comparison of frequencies by logistic regression method, including age, sex, and TI as covariates. We considered a significant association when *p* < 0.05. GF: genotype frequency, %; OR: odds ratio; CI: confidence interval; SNP: single nucleotide polymorphism; Ref: reference.

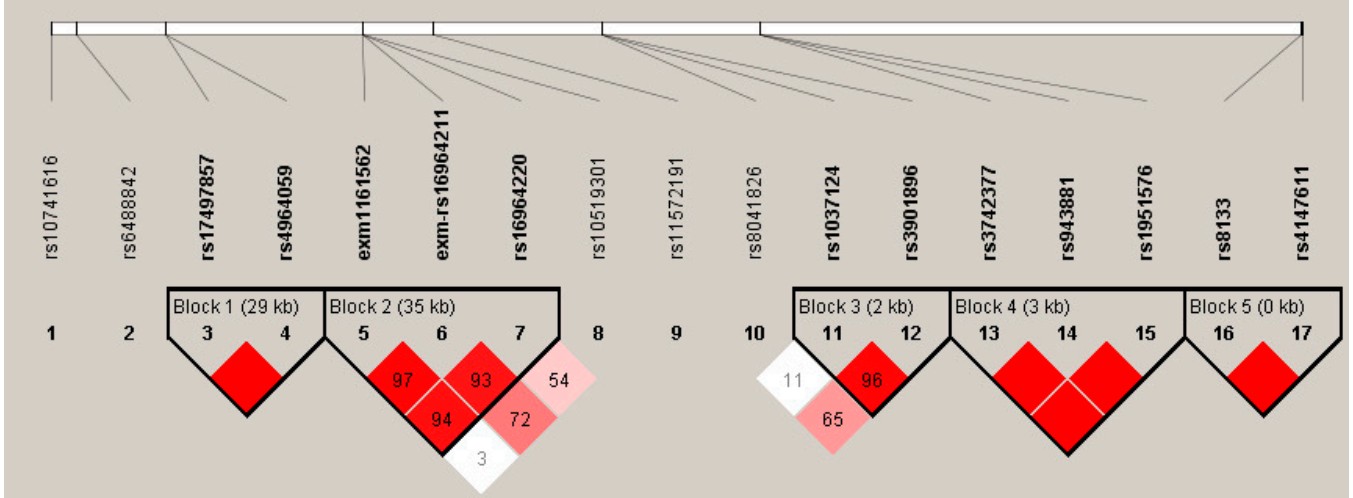

**Figure 2.** Haplotype block for SNPs associated in the smokers' comparison. Five blocks with LD > 80 were predicted, block 1 is conformed of two SNPs in *ARNTL2*, block 2 by 3 SNPs in *CYP19A1*, block 3 by SNPs in *ARNT2*, block 4 with 3 SNPs in *CYP46A1*, and block 5 by 3 SNPs in *MGST3*. Color and values in the haplotypes represent D' LD values. The color intensity corresponds to the higher LD value.

**Table 4.** Haplotypes association analysis in smokers' comparison.

| Haplotypes | COPD-S (n = 141) Freq% | SWOC (n = 213) Freq% | *p* | OR | CI (95%) |
|---|---|---|---|---|---|
| | | rs10741616-rs7126796 (*ARNTL*) | | | |
| GT | 49.8 | 57.5 | 0.046 | 0.74 | (0.55–0.99) |
| | | rs11048977-rs1037924-rs17497857-rs7138982 (*ARNTL2*) | | | |
| GACC | 12.8 | 8.0 | 0.037 | 1.69 | (1.03–2.77) |
| | | rs10046-rs700519-rs6493489-rs2899472-rs2414095-rs700518 (*CYP19A1*) | | | |
| CACCGA | 3.5 | 7.7 | 0.022 | 0.44 | (0.21–0.9) |
| | | rs1374213-rs3901896-rs7168908-rs2278709 (*ARNT2*) | | | |
| TTGC | 48.6 | 57 | 0.027 | 0.71 | (0.53–0.96) |
| TCGC | 14.5 | 8.7 | 0.015 | 1.79 | (1.12–2.87) |
| | | rs3742377-rs943881-rs1951576-rs12435918-rs2146238 (*CYP46A1*) | | | |
| AAAAG | 14.9 | 21.1 | 0.037 | 0.65 | (0.44–0.98) |
| | | rs10847-rs11552229-rs2228099 (*ARNT*) | | | |
| CAC | 30.5 | 40.5 | 0.007 | 0.65 | (0.44–0.89) |
| | | rs8133-rs4147611 (*MGST3*) | | | |
| GT | 42.2 | 51.9 | 0.012 | 0.68 | (0.5–0.92) |
| TG | 30.1 | 21.4 | 0.008 | 1.59 | (1.13–2.24) |

Haplotypes association analysis corrected by covariates (age, sex, and BEI). Data are presented as% frequency. $\chi^2$ was carried out to calculate *p*-values, OR and CI (95%); we considered significant association when *p* < 0.05. Freq%: Frequency in %; OR: odds ratio; CI: confidence interval.

### 3.3. Severity Analysis

We stratify COPD-S and COPD-BBS subjects according to the GOLD stages (mild stages: GOLD I + II; severe stages: GOLD III + IV). In COPD-S, we found four SNPs: rs12435918 in CYP46A1, rs625456 in GSTM2, and rs1058930 in CYP2C8 associated with severe forms of COPD secondary to tobacco smoking (Supplementary Table S8). For COPD-BBS, we found rs12300289 in *ARNTL2*, rs10847 in *ARNT*, and rs2234696 in *GSTM3* associated with the severe form of COPD secondary to biomass-burning smoke exposition (Supplementary Table S9).

### 3.4. Association Analysis in the Group Exposed to BBS

In the BBS exposure comparison group (COPD-BBS vs. BBES), 24 SNPs were found to be significantly associated (*p* < 0.05), of which twenty were associated with a higher risk of COPD and four with a decreased risk of suffering from the disease. Interestingly, the associated polymorphisms are mainly distributed in the *MGST3, MGST1, CYP2C8*, and *CYP2C9* genes (Table 5). After applying the Bonferroni correction test, only three SNPs remained associated, rs11799886/*MGST3* (*p* = 0.019), rs1856908/*CYP2C9* (*p* = 0.003), and rs1934953/*CYP2C8* (*p* = 0.021).

When performing the genotype analysis, 23 SNPs associated with the disease were found; three with reduced risk and twenty with a higher COPD risk. In six SNPs, no homozygotes were found for the minor allele and the leading associations were with the heterozygous genotypes. It should be noted that the groups of SNPs in the *MGST3, CYP2C8, CYP2C9*, and *MGST1* genes remained associated. *MGST3* presented the highest number of associated SNPs and OR values, presenting a three-fold increased risk of developing the disease. In the case of *CYP2C8*, although only three SNPs were found to be associated with increased risk, their OR values were also up to four times higher risk of developing COPD secondary to BBS (Table 6).

**Table 5.** Allele association analysis in exposed biomass-burning smoke comparison.

| SNP/Allele | COPD-BBS (n = 98) | AF% | BBES (n = 293) | AF% | OR | CI (95%) | *p* | *p* * |
|---|---|---|---|---|---|---|---|---|
| | | | rs4147611/*MGST3* | | | | | |
| T | 125 | 63.78 | 450 | 76.79 | | 1.00 (Ref.) | | |
| G | 71 | 36.22 | 136 | 23.21 | 1.94 | 1.33–2.85 | 0.0007 | NS |
| | | | rs11799886/*MGST3* | | | | | |
| G | 16 | 8.16 | 12 | 2.05 | | 1.00 (Ref.) | | |
| A | 180 | 91.84 | 574 | 97.95 | 3.96 | 1.66–9.49 | 0.002 | 0.019 |
| | | | rs6681/*MGST3* | | | | | |
| C | 190 | 96.94 | 582 | 99.32 | | 1.00 (Ref.) | | |
| T | 6 | 3.06 | 4 | 0.68 | 8.99 | 1.85–43.78 | 0.007 | NS |
| | | | rs9333378/*MGST3* | | | | | |
| A | 149 | 76.02 | 497 | 84.81 | | 1.00 (Ref.) | | |
| G | 47 | 23.98 | 89 | 15.19 | 1.72 | 1.12–2.64 | 0.013 | NS |
| | | | rs957644/*MGST3* | | | | | |
| C | 176 | 89.80 | 563 | 96.08 | | 1.00 (Ref.) | | |
| T | 20 | 10.20 | 23 | 3.92 | 2.26 | 1.17– 4.37 | 0.015 | NS |
| | | | rs10789501/*CYP4A22* | | | | | |
| C | 90 | 45.92 | 308 | 52.56 | | 1.00 (Ref.) | | |
| T | 106 | 54.08 | 278 | 47.44 | 1.58 | 1.07–2.34 | 0.021 | NS |
| | | | rs6690005/*CYP4Z1* | | | | | |
| A | 92 | 46.94 | 306 | 52.22 | | 1.00 (Ref.) | | |
| G | 104 | 53.06 | 280 | 47.78 | 1.55 | 1.06–2.29 | 0.026 | NS |
| | | | rs12059860/*CYP4B1* | | | | | |
| T | 186 | 94.90 | 577 | 98.46 | | 1.00 (Ref.) | | |
| C | 10 | 5.10 | 9 | 1.54 | 15.06 | 1.38–164 | 0.026 | NS |
| | | | rs1856908/*CYP2C9* | | | | | |
| T | 139 | 70.92 | 498 | 84.98 | | 1.00 (Ref.) | | |
| G | 57 | 29.08 | 88 | 15.02 | 2.05 | 1.31–3.19 | 0.002 | 0.003 |
| | | | rs1934953/*CYP2C8* | | | | | |
| G | 135 | 68.88 | 482 | 82.25 | | 1.00 (Ref.) | | |
| A | 61 | 31.12 | 104 | 17.75 | 2.01 | 1.29–3.12 | 0.002 | 0.021 |
| | | | rs3752988/*CYP2C8* | | | | | |
| T | 160 | 81.63 | 530 | 90.44 | | 1.00 (Ref.) | | |
| C | 36 | 18.37 | 56 | 9.56 | 2.06 | 1.19–3.57 | 0.01 | NS |
| | | | rs9332220/*CYP2C9* | | | | | |
| G | 173 | 88.27 | 558 | 95.22 | | 1.00 (Ref.) | | |
| A | 23 | 11.73 | 28 | 4.78 | 2.29 | 1.15–4.56 | 0.019 | NS |
| | | | rs1801253/*ADRB1* | | | | | |
| C | 179 | 91.33 | 568 | 96.93 | | 1.00 (Ref.) | | |
| G | 17 | 8.67 | 18 | 3.07 | 2.49 | 1.13–5.53 | 0.024 | NS |
| | | | rs10509681/*CYP2C8* | | | | | |
| T | 184 | 93.88 | 575 | 98.12 | | 1.00 (Ref.) | | |
| C | 12 | 6.12 | 11 | 1.88 | 2.73 | 1.09–6.86 | 0.033 | NS |
| | | | rs12794714/*CYP2R1* | | | | | |
| G | 118 | 60.20 | 298 | 50.85 | | 1.00 (Ref.) | | |
| A | 78 | 39.80 | 288 | 49.15 | 0.54 | 0.36–0.81 | 0.0026 | NS |
| | | | rs1138272/*GSTP1* | | | | | |
| C | 192 | 97.96 | 583 | 99.49 | | 1.00 (Ref.) | | |
| T | 4 | 2.04 | 3 | 0.51 | 8.95 | 1.563–51.22 | 0.014 | NS |
| | | | rs7129781/*CYP2R1* | | | | | |
| T | 186 | 94.90 | 576 | 98.29 | | 1.00 (Ref.) | | |
| C | 10 | 5.10 | 10 | 1.71 | 2.97 | 1.039– 8.49 | 0.042 | NS |
| | | | rs1913263/*MGST1* | | | | | |
| G | 90 | 45.92 | 350 | 59.73 | | 1.00 (Ref.) | | |
| A | 106 | 54.08 | 236 | 40.27 | 1.86 | 1.26–2.735 | 0.002 | NS |

**Table 5.** *Cont.*

| SNP/Allele | COPD-BBS (n = 98) | AF% | BBES (n = 293) | AF% | OR | CI (95%) | *p* | *p* * |
|---|---|---|---|---|---|---|---|---|
| | | | rs1042669/*MGST1* | | | | | |
| T | 147 | 75.00 | 380 | 64.85 | | 1.00 (Ref.) | | |
| G | 49 | 25.00 | 206 | 35.15 | 0.61 | 0.39–0.94 | 0.024 | NS |
| | | | rs9332959/*MGST1* | | | | | |
| G | 147 | 75.00 | 381 | 65.02 | | 1.00 (Ref.) | | |
| T | 49 | 25.00 | 205 | 34.98 | 0.63 | 0.41–0.96 | 0.031 | NS |
| | | | rs4149197/*MGST1* | | | | | |
| G | 115 | 58.67 | 399 | 68.09 | | 1.00 (Ref.) | | |
| C | 81 | 41.33 | 187 | 31.91 | 1.52 | 1.01–2.28 | 0.044 | NS |
| | | | rs11048977/*ARNTL2* | | | | | |
| G | 151 | 77.04 | 409 | 69.80 | | 1.00 (Ref.) | | |
| A | 45 | 22.96 | 177 | 30.20 | 0.64 | 0.48–0.99 | 0.047 | NS |
| | | | rs2899472/*CYP19A1* | | | | | |
| C | 187 | 95.41 | 573 | 97.78 | | 1.00 (Ref.) | | |
| A | 9 | 4.59 | 13 | 2.22 | 2.9 | 1.09–7.72 | 0.033 | NS |
| | | | rs117987520/*CYP11A1* | | | | | |
| G | 193 | 98.47 | 585 | 99.83 | | 1.00 (Ref.) | | |
| A | 3 | 1.53 | 1 | 0.17 | 11.67 | 1.08–126.5 | 0.043 | NS |

Comparison of frequencies by logistic regression method, including age and BEI as covariates. We considered a significant association when *p* < 0.05. * *p*-value after Bonferroni correction test, AF: allele frequency, %. OR: odds ratio; CI: confidence interval; SNP: single nucleotide polymorphism; Ref: reference; NS: no significant.

For BBS comparison, we included 298 SNPs after filtering by MAF (>1%). We did not find any clusters with more than 2% of the variance (Supplementary Figure S4B). Looking for other clustering patterns, we included only the SNPs associated with COPD-BBS, but no grouping patterns that could explain higher variability were found (<1%) (Supplementary Figure S6).

We found seven blocks of haplotypes in high LD in the genes *ARNTL*, *CYP2R1*, *MGST1*, *ARNTL2*, *GSTP1*, *CYP1A2*, *ARNT2*, *CYP2C18*, *CYP2C9*, *CYP2C8*, *GSTM5*, *GSTM3*, and *MGST3* (Figure 3). In block 4, we found the rs1856908 reported in allele and genotype analysis. A haplotype block (block 7) was found in MGST3; this block was found in the smokers' comparison (rs8133-rs4147611).

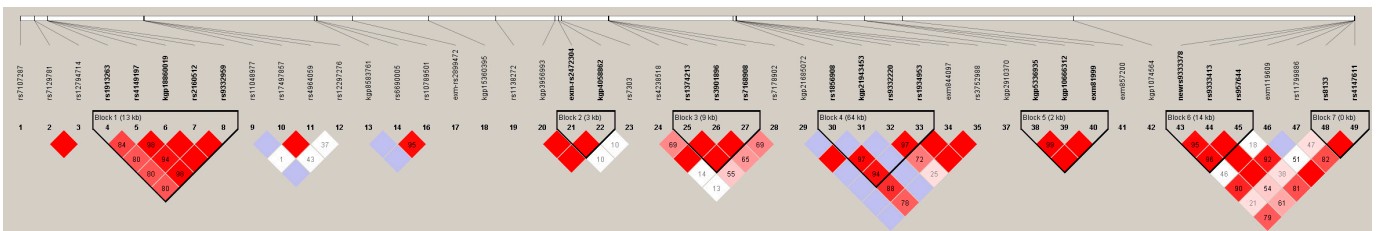

**Figure 3.** Haplotypes blocks include all the SNPs associated with BBS comparison. Seven blocks were identified in the biomass comparison. Color and values in the diamonds represent D' LD values. The intensity of the color is directly proportional to the higher LD value.

Eight combinations of SNPs were associated with a lower risk of suffering COPD (OR < 1) and eighteen were associated with a higher risk (OR > 1.5). The larger SNP combination was composed of 15 variants that range from *CYP2C18* to *CYP2C9*, with the highest OR value at almost eight times higher risk of COPD. *MGST3* was the gene with more blocks; we found three haplotypes, and the SNPs included in the haplotypes had been previously reported in alleles and genotypes analyses (Table 7).

**Table 6.** Genotype association analysis in the exposed to biomass-burning smoke.

| SNP/Allele | COPD-BBS (n = 98) | GF% | BBES (n = 293) | GF% | OR | CI (95%) | *p* |
|---|---|---|---|---|---|---|---|
| | | | rs12059860/*CYP4B1* | | | | |
| TT | 88 | 89.80 | 284 | 96.93 | | 1.00 (Ref.) | |
| TG | 10 | 10.20 | 9 | 3.07 | 15.44 | 1.79–335.6 | 0.025 |
| GG | 0 | 0 | 0 | 0.00 | NA | NA | NA |
| | | | rs6690005/*CYP4Z1* | | | | |
| AA | 21 | 21.43 | 81 | 27.65 | | 1.00 (Ref.) | |
| AG | 50 | 51.02 | 144 | 49.15 | 1.65 | 0.81–3.45 | 0.17 |
| GG | 27 | 27.55 | 68 | 23.21 | 2.86 | 1.26–6.69 | 0.013 |
| | | | rs10789501/*CYP4A22* | | | | |
| CC | 19 | 19.39 | 82 | 27.99 | | 1.00 (Ref.) | |
| CT | 52 | 53.06 | 144 | 49.15 | 1.75 | 0.87–3.68 | 0.13 |
| TT | 27 | 27.55 | 67 | 22.87 | 2.84 | 1.24–6.73 | 0.015 |
| | | | rs9333378/*MGST3* | | | | |
| AA | 59 | 60.20 | 214 | 73.04 | | 1.00 (Ref.) | |
| AG | 31 | 31.63 | 69 | 23.55 | 1.63 | 0.98–2.72 | 0.041 |
| GG | 8 | 8.16 | 10 | 3.41 | 2.90 | 1.09–7.68 | 0.027 |
| | | | rs9333413/*MGST3* | | | | |
| AA | 39 | 39.80 | 120 | 40.96 | | 1.00 (Ref.) | |
| AG | 36 | 36.73 | 136 | 46.42 | 0.91 | 0.48–1.72 | 0.77 |
| GG | 23 | 23.47 | 36 | 12.29 | 2.22 | 1.03–4.79 | 0.042 |
| | | | rs957644/*MGST3* | | | | |
| CC | 80 | 81.63 | 271 | 92.49 | | 1.00 (Ref.) | |
| CT | 16 | 16.33 | 21 | 7.17 | 2.29 | 1.02–5.07 | 0.041 |
| TT | 2 | 2.04 | 1 | 0.34 | 5.26 | 0.48–116.3 | 0.18 |
| | | | rs6681/*MGST3* | | | | |
| CC | 92 | 93.88 | 289 | 98.63 | | 1.00 (Ref.) | |
| CT | 6 | 6.12 | 4 | 1.37 | 9.77 | 2.11–54.79 | 0.005 |
| TT | 0 | 0.00 | 0 | 0.00 | NA | NA | NA |
| | | | rs11799886/*MGST3* | | | | |
| GG | 82 | 83.67 | 281 | 95.90 | | 1.00 (Ref.) | |
| GA | 16 | 16.33 | 12 | 4.10 | 4.51 | 1.82–11.47 | 0.001 |
| AA | 0 | 0.00 | 0 | 0.00 | NA | NA | NA |
| | | | rs8133/*MGST3* | | | | |
| GG | 74 | 75.51 | 243 | 82.94 | | 1.00 (Ref.) | |
| GT | 19 | 19.39 | 48 | 16.38 | 1.32 | 0.63–2.67 | 0.45 |
| TT | 5 | 5.10 | 2 | 0.68 | 12.44 | 2.31–99.26 | 0.006 |
| | | | rs4147611/*MGST3* | | | | |
| TT | 43 | 43.88 | 179 | 61.09 | | 1.00 (Ref.) | |
| TG | 39 | 39.80 | 92 | 31.40 | 2.04 | 1.09–3.80 | 0.025 |
| GG | 16 | 16.33 | 22 | 7.51 | 4.57 | 1.88–11.21 | 0.0008 |
| | | | rs1856908/*CYP2C9* | | | | |
| TT | 50 | 51.02 | 213 | 72.70 | | 1.00 (Ref.) | |
| TG | 39 | 39.80 | 72 | 24.57 | 2.51 | 1.36–4.67 | 0.003 |
| GG | 9 | 9.18 | 8 | 2.73 | 4.59 | 1.43–14.88 | 0.009 |
| | | | rs9332220/*CYP2C9* | | | | |
| GG | 77 | 78.57 | 266 | 90.78 | | 1.00 (Ref.) | |
| GA | 19 | 19.39 | 26 | 8.87 | 2.41 | 1.07–5.33 | 0.031 |
| AA | 2 | 2.04 | 1 | 0.34 | 2.57 | 0.09–66.83 | 0.51 |
| | | | rs1934953/*CYP2C8* | | | | |
| GG | 47 | 47.96 | 198 | 67.58 | | 1.00 (Ref.) | |
| GA | 41 | 41.84 | 86 | 29.35 | 2.53 | 1.38–4.68 | 0.0027 |
| AA | 10 | 10.20 | 9 | 3.07 | 4.93 | 1.54–16.18 | 0.0072 |

**Table 6.** *Cont.*

| SNP/Allele | COPD-BBS (n = 98) | GF% | BBES (n = 293) | GF% | OR | CI (95%) | *p* |
|---|---|---|---|---|---|---|---|
| | | | rs10509681/*CYP2C8* | | | | |
| TT | 87 | 88.78 | 283 | 96.59 | | 1.00 (Ref.) | |
| TC | 10 | 10.20 | 9 | 3.07 | 4.69 | 1.48–14.87 | 0.007 |
| CC | 1 | 1.02 | 1 | 0.34 | 3.13 | 0.12–84.55 | 0.44 |
| | | | rs3752988/*CYP2C8* | | | | |
| TT | 64 | 65.31 | 240 | 81.91 | | 1.00 (Ref.) | |
| TC | 32 | 32.65 | 50 | 17.06 | 2.98 | 1.54–5.78 | 0.001 |
| CC | 2 | 2.04 | 3 | 1.02 | 4.5 | 0.45–45.74 | 0.18 |
| | | | rs1801253/*ADRB1* | | | | |
| CC | 81 | 82.65 | 276 | 94.20 | | 1.00 (Ref.) | |
| CG | 17 | 17.35 | 16 | 5.46 | 3.01 | 1.24–7.24 | 0.014 |
| GG | 0 | 0.00 | 1 | 0.34 | NA | NA | NA |
| | | | rs12794714/*CYP2R1* | | | | |
| GG | 36 | 36.73 | 76 | 25.94 | | 1.00 (Ref.) | |
| GA | 46 | 46.94 | 146 | 49.83 | 0.47 | 0.25–0.89 | 0.022 |
| AA | 16 | 16.33 | 71 | 24.23 | 0.35 | 0.14–0.79 | 0.015 |
| | | | rs1913263/*MGST1* | | | | |
| GG | 22 | 22.45 | 100 | 34.13 | | 1.00 (Ref.) | |
| GA | 46 | 46.94 | 150 | 51.19 | 1.41 | 0.71–2.91 | 0.33 |
| AA | 30 | 30.61 | 43 | 14.68 | 3.42 | 1.57–7.68 | 0.002 |
| | | | rs4149197/*MGST1* | | | | |
| GG | 35 | 35.71 | 129 | 44.03 | | 1.00 (Ref.) | |
| GC | 45 | 45.92 | 141 | 48.12 | 0.96 | 0.51–1.79 | 0.89 |
| CC | 18 | 18.37 | 23 | 7.85 | 3.73 | 1.56–9.06 | 0.003 |
| | | | rs1042669/*MGST1* | | | | |
| TT | 56 | 57.14 | 123 | 41.98 | | 1.00 (Ref.) | |
| TG | 35 | 35.71 | 134 | 45.73 | 0.44 | 0.24–0.79 | 0.007 |
| GG | 7 | 7.14 | 36 | 12.29 | 0.5 | 0.17–1.32 | 0.18 |
| | | | rs9332959/*MGST1* | | | | |
| GG | 56 | 57.14 | 124 | 42.32 | | 1.00 (Ref.) | |
| GT | 35 | 35.71 | 133 | 45.39 | 0.46 | 0.25–0.83 | 0.01 |
| TT | 7 | 7.14 | 36 | 12.29 | 0.52 | 0.17–1.34 | 0.19 |
| | | | rs2899472/*CYP19A1* | | | | |
| CC | 89 | 90.82 | 280 | 95.56 | | 1.00 (Ref.) | |
| CA | 9 | 9.18 | 13 | 4.44 | 3.2 | 1.17–8.57 | 0.021 |
| AA | 0 | 0.00 | 0 | 0.00 | NA | NA | NA |
| | | | rs117987520/*CYP11A1* | | | | |
| GG | 95 | 96.94 | 292 | 99.66 | | 1.00 (Ref.) | |
| GA | 3 | 3.06 | 1 | 0.34 | 12.9 | 1.45–2.8 | 0.036 |
| AA | 0 | 0.00 | 0 | 0 | NA | NA | NA |

Comparison of frequencies by logistic regression method, including age and BEI as covariates. We considered a significant association when *p* < 0.05. GF: genotype frequency; OR = odds ratio; CI = confidence interval; SNP = singles nucleotide polymorphism; Ref = reference; NA = not applicable.

In the analysis by severity, we found four SNPs associated, three with the severe GOLD stages and one with mild COPD stages. When data were corrected by covariates, three out of four SNPs remained associated. However, no SNP retained its association after Bonferroni correction. (Supplementary Table S8). We found five SNPs in the severity analysis of the COPD-BBS group, four associated with a severe form of COPD and one with a mild form of the illness. Although four remained associated after the correction by covariates, no SNP conserves association after Bonferroni adjustment (Supplementary Table S9).

**Table 7.** Haplotypes association analysis in BBS comparison.

| Haplotypes | COPD-BBS (n = 98) Freq% | BBES (n = 293) Freq% | $p$ | OR | CI (95%) |
|---|---|---|---|---|---|
| | | rs10741616-rs7126796 (*ARNTL*) | | | |
| AT | 28.3 | 20.6 | 0.023 | 1.51 | (1.05–2.18) |
| | | rs1993116-rs12794714 (*CYP2R1*) | | | |
| CA | 40.1 | 49.3 | 0.024 | 0.69 | (0.49–0.95) |
| CG | 18.3 | 12 | 0.023 | 1.65 | (1.07–2.54) |
| | | rs1913263-rs4149192 (*MGST1*) | | | |
| GG | 45.5 | 59.7 | $5 \times 10^{-4}$ | 0.56 | (0.41–0.78) |
| AG | 43.1 | 32.6 | 0.007 | 1.56 | (1.13–2.17) |
| | | rs7312090-rs11875-rs1042669-rs2160512-rs9332959-rs6488842 (*MGST1*) | | | |
| GGGGTC | 24.8 | 35.1 | 0.007 | 0.61 | (0.42–0.87) |
| GGTAGC | 25.5 | 18.4 | 0.032 | 1.53 | (1.05–2.23) |
| | | rs1037924-rs17497857-rs7138982-rs6487604-rs4964059 (*ARNTL2*) | | | |
| ACCCC | 9.9 | 4.6 | 0.006 | 2.28 | (1.25–4.15) |
| | | rs1695-rs4891 (*GSTP1*) | | | |
| AC | 0.4 | 0.1 | 0.007 | 3.98 | (1.37–11.63) |
| | | rs2472304-rs2470890 (*CYP1A2*) | | | |
| GC | 84.7 | 91.1 | 0.009 | 0.54 | (0.33–0.87) |
| AT | 15.3 | 8.9 | 0.009 | 1.86 | (1.16–2.99) |
| | | rs1374213-rs3901896-rs7168908-rs2278709 (*ARNT2*) | | | |
| TTGC | 68.3 | 76.5 | 0.022 | 0.66 | (0.47–0.94) |
| CCAC | 6.9 | 1.4 | $3.5 \times 10^{-5}$ | 5.38 | (2.22–13.02) |
| | | rs11856676-rs4238522 (*ARNT2*) | | | |
| TT | 10.4 | 5.8 | 0.027 | 1.88 | (1.07–3.33) |
| | | rs2901783-rs76498052-rs1126545-rs2860840-rs1042192-rs1042194-rs7916649-rs4388808-rs4244285-rs12767583-rs4494250-rs1853205-rs10786172-rs28399505-rs1856908 (*CYP2C18, CYP2C9*) | | | |
| ACCTGGGAGCAGGTT | 38.7 | 49.2 | 0.01 | 0.65 | (0.46–0.9) |
| ACCTGGGAGCAGGTG | 5.8 | 0.9 | $3.9 \times 10^{-5}$ | 7.34 | (2.55–21.09) |
| ACCCGGGAGCGGATG | 3 | 0.5 | 0.005 | 5.94 | (1.47–24.01) |
| | | rs1058932-rs11572177-rs1934953-rs1934951-rs11572101-rs11572093-rs3752988-rs1934956 (*CYP2C8*) | | | |
| CAAGTGCC | 8.9 | 3.2 | 0.001 | 2.92 | (1.5–5.68) |
| | | rs11807-rs1055259-rs3814309 (*GSTM5/GSTM3*) | | | |
| ATT | 38.9 | 28.7 | 0.007 | 1.59 | (1.14–2.23) |
| | | rs1537236-rs7483 (*GSTM3*) | | | |
| TT | 46.9 | 55.1 | 0.044 | 0.72 | (0.52–0.99) |
| CC | 38.6 | 28.2 | 0.006 | 1.61 | (1.15–2.25) |
| | | rs4147592-rs4147594-rs4147595 (*MGST3*) | | | |
| GCC | 10.9 | 5.3 | 0.006 | 2.18 | (1.23–3.86) |
| | | rs9333413-rs957644 (*MGST3*) | | | |
| GT | 9.9 | 3.9 | 0.001 | 2.69 | (1.44–5.01) |
| | | rs8133-rs4147611 (*MGST3*) | | | |
| GT | 63.9 | 76.8 | $3 \times 10^{-4}$ | 0.53 | (0.38–0.75) |
| GG | 21.8 | 14.3 | 0.013 | 1.66 | (1.11–2.49) |
| TG | 14.4 | 8.9 | 0.027 | 1.72 | (1.06–2.79) |

Haplotypes association analysis corrected by covariates (age, sex, and BEI). Data are presented as% frequency. $\chi^2$ was carried out to calculate *p*-values, OR and CI (95%); we considered significant association when $p < 0.05$. Freq%: frequency in percentages; OR: odds ratio; CI: confidence interval.

## 4. Discussion

Although multiple GWAS have described associations with COPD, most studies focus on COPD secondary to tobacco smoking in Caucasian populations from Europe and the USA; we analyzed SNPs in exome regions in the whole human genome by the array genotyping technology looking for variants associated with COPD both secondary to tobacco smoking or biomass-burning smoke in the Mexican mestizo population. The participants were recruited from different campaigns of COPD early detection in Mexico City and the highlands of Oaxaca.

In our group, Perez-Rubio et al. had previously described the genetic component of the population included in this study, demonstrating the contribution of the Amerindian/Caucasian genetic component [29]. All patients had at least three prior generations born in Mexico (parents and grandparents) and were considered Mexican mestizos. We have previously demonstrated that this criterion is a good proxy of Mexican ancestry evaluated by ancestry-informative markers [30].

We found differences in variables, such as sex, age, BMI, and tobacco index, in comparing smokers. Due to these differences, we included these covariates in the association of alleles and haplotypes analyses to avoid false positive findings. For the BBS group, we found differences in age and exposure data. We did not find differences in the men/women ratio, but women are predominantly represented in both groups. Low- to middle-income countries are the principal users of biomass, and each region worldwide reported the use of specific kind of biomass; for example, in China, there is a predominance in the use of charcoal and coal; in Nepal and Kenya, the use of manure from big ruminants is a common practice; in a large variety of Latin America, African and South Asian countries is predominant the use of firewood from a great variety of trees and even agriculture waste [18]. The primary biomass fuel used in Mexico is firewood or mixtures of firewood, manure, and farming waste, especially in rural or suburban areas. The principal population exposed are women and children because women are the principal family members in charge of cooking [18,31].

Rehfuess and collaborators establish that 52% of the world population uses either biomass or solid fuel. Stratifying six geographic areas, they determined that Africa, South Asia, and different areas of Latin America are the principal biomass users [32].

The World Health Organization reported that around 2.5 billion people used any biomass only to cook, and, especially in rural zones, combustion takes place indoors, in closed or poorly ventilated places using improvised stoves or pipes, resulting in an event called "indoor pollution", affecting mainly women and children, and producing 1.3 million of premature deaths associated with respiratory diseases and infections [31,33].

Candidate genes analysis methodologies are strategies for post-genotyping data in genome-wide studies (GWAS) [34]. In this study, we used genotyping exome array that includes up to 560 thousand specific sequence probes capable of detecting the SNPs in exome regions. We included genes whose biological function was related to xenobiotic and drug metabolism processing.

Xenobiotics are exogenous bodily substances that involve absorption, distribution, and metabolism [35]. Some genes related to the xenobiotic processing are genes of the glutathione transferase family (phase 2 metabolizers) [36], cytochrome P450 (phase 1 metabolizing isoenzymes) [37], aryl hydro-carbon receptors and translocators [38], and ADRB1 genes (β1 adrenergic receptors) [39].

The genes included were *CYP4B1, CYP4Z2P, CYP4A11, CYP4 × 1, CYP4Z1, CYP4A22, CYP2J2, CYP26C1, CYP26A1, CYP2C18, CYP2C19, CYP2C9, CYP2C8, CYP17A1, CYP2E1, CYP2R1, CYP27B1, ACYP1, CYP46A1, CYP19A1, CYP11A1, CYP1A1, CYP1A2, GSTM4, GSTM3, GSTM2, GSTM1, MGST3, GSTO1, GSTO2, GSTP1, MGST1, GSTZ1, ADRB1, ARNT2, ARNT, ARNTL2,* and *ARNTL,* and a total of 750 SNPs were selected.

In smokers' comparison, we found eight SNPs associated with COPD; six SNPs were associated with a higher risk of suffering COPD-S; rs11572191 in *CYP2J2*; rs8133 in *MGST3*; rs17497857 and rs4964059 in *ARNTL2*; and rs3901896 and rs8041826 in *ARNT2*, all with the

minor allele. Only two SNPs were found associated with lower risk; rs4147611 in *MGST3* and rs3742377 in *CYP46A1*.

Our is the first study reporting these sets of SNPs with COPD-S, particularly in a mestizo (admixed) population as the Mexican. Although any polymorphism in our findings was previously described, the genes associated are reported in different studies as associated factors to lung diseases. Four SNPs in *CYP2J2* were found to be associated with the Chinese Han population with COPD-S [40], and even in the Russian population, SNPs in *CYP2J2* are associated with bronchitis secondary to smoking [41]. Other investigations have demonstrated that SNPs in *CYP2J2* could be involved in lung ischemia and reperfusion injury, especially in smokers [42,43]. Although our investigation focuses on COPD, lung injury and hypertension are common in subjects with COPD. Additionally, *CYP2J2* is related to asthma models and cancer. Refs. [44–46] *CYP2C9* has been included in studies related to adenocarcinoma and other forms of lung cancer [45,47,48].

Even though there are few reports of *MGST3* and COPD, some SNPs have been associated with attenuating smokers' accelerated decline in FEV1/FVC [49]. No other reports of lung diseases have been reported.

ARNT genes encode proteins capable of binding to aryl hydro-carbon receptors to translocate them to the cell nucleus as transcription factors related to gene promoters such as HIF1α. The principal studies between ARNT genes suggest a possible relation with small-cell cancer [50–53].

The protein encoded by *MGST1* (microsomal glutathione S-transferase 1) is a membrane-associated protein with peroxidase activity which avoids lipid damage against reactive oxygen species (ROS), cytotoxic, and drugs. The principal association between *MGST1* and lung diseases includes different types of cancer, such as adenocarcinoma or non-small cell lung cancer [54,55]. Woldhuis et al. proposed that microsomal glutathione S-transferase 1 could be related to cell senescence and extracellular matrix reorganization [56]. Recently, ferroptosis has been described as a programmed dead type with a higher lipid peroxide concentration in other illnesses. *MGST1* is differentially expressed in alveolar type 2 cells [57]. In the case of *MGST3*, sets of polymorphism attenuated lung function decline in European-American smokers [49].

In genotypes, only higher risk associated SNPs were found associated with the illness; among these, four were previously described in allele analysis; rs11572191, rs17497857, rs3901896, and rs8041826. Three more SNPs were found in genotype analysis, the GG of rs1951576 and CC genotype of rs943881 in gene *CYP46A1* and TT of rs6488842 in gene *MGST1*. The alleles associated with a low risk of COPD were possibly not found in the genotype phase due to the low frequency of minor alleles; not enough homozygous minor allele genotypes were found.

There is limited information regarding the severity data related to the SNPs and genes associated with xenobiotic metabolism. Studies in emphysema have demonstrated that the expression of *GSTM3* was upregulated in mild illness [58], while other studies describe SNPs associated with a lower FEV1/FVC ratio [59]. *GSTM3* is a gene in which protein product is related to eliminating electrophilic compounds and carcinogens. We found rs2234696 in *GSTM3* to be associated with the severe form of COPD in smokers, and while there are no reports regarding the SNP, we can state that the SNP could affect the structure of the protein codified by the gene, thus preventing its biological function.

Haplotypes analysis is used to elucidate possible associations in groups of SNPs in different regions of genes [60]. For the comparison with smokers, we found five blocks with high LD (>85) in *ARNTL1*, *CYP19A1*, *ARNT2*, *CYP46A1*, and *MGST3*. Multiple genes have been associated with complex diseases like COPD but with moderate OR [61]. Including multiple analyses as polygenic risk scores has demonstrated that a combination of genetic variants could explain the multiples association and even reach the haplotype analysis [62]. In the haplotype blocks, we found five combinations of SNPs associated with allele analysis, suggesting a probably critical role in COPD pathophysiology.

We found more SNPs associated with COPD-BBS than COPD-S; at the allele level, the principal findings include SNPs in *MGST3*, *CYP2C8*, and *MGST1*. Few studies have been made in the genetic field about COPD-BBS. Our current study is the first in exome-wide genotyping.

Principal studies with COPD-BBS in Latin America emphasize clinical description; other studies include Chinese and Chilean populations but focus on genes such as *PRDM15* and *CXCL10*, respectively [63,64]. Additionally, we found a greater number of SNPs in COPD-BBS than in COPD-S, which could suggest a possible major complex in developing COPD-BBS.

In haplotype analysis, we found seven blocks in high LD; among these findings, the larger haplotype block was found in *MGST1*, and the leading role of the protein encoded is related to extracellular matrix reorganization. A clinical characteristic of COPD-BBS is anthracofibrosis, bronchial caliber diminution, and increased mucus production [65]. This bronchial remodeling could be related to genes such as *MGST1*, but we cannot demonstrate it due to the limited investigation of the cellular effects of the BBS.

Additionally, we found two different haplotypes in *ARNT2* associated higher risk of COPD-BBS. Previous reports about COPD focus on tobacco smoking, and some of the most significant results involve AHR and ARNT family genes. The evidence demonstrates an important role of AHR in attenuating inflammation related to neutrophils [66] and in lung remodeling by different genes such as *MMP9* [67]. Studies in animal models have demonstrated a possible relationship between the aryl hydrocarbon receptors and CYP genes, especially in asthma, which control inflammatory processes [68]. We found many haplotypes in CYP and ARNT genes, which could support the biological relation.

Even though we found SNPs associated specifically with COPD-S or COPD-BBS, we also found similar SNPs and haplotypes, such as *ARNT2* and *MGST3*. This result could suggest the participation of a molecular shared component. Using in silico databases, such as GTEx, we found four SNPs (rs6681 and rs9333378 in *MGST3*, rs10789501 in *CYP4A22*, rs117987520 in *CYP11A1*) that affect the expression levels in the genes where they are located.

With the MCA, we included the SNPs associated with each subtype of COPD, but we did not find clear subgroups. Some studies have demonstrated that multivariate analysis as MCA and polygenic risk score calculation could give more information regarding the effect of exposure/clinical variables and genetic variants as SNPs [69].

Other phenomena reported in our investigation are the SNPs associated with a lower risk of COPD. In previous investigations, we have described similar associations with other SNPs in different genes [70,71]. This effect is described in different illnesses, called the "Hispanic Paradox", a theory that describes the role of the genetic background of Amerindians which could lead to lower severity or better prognosis in illnesses, including COPD [72,73].

Our is the first exome-wide association study in Mexican mestizos with COPD, classified by tobacco smoking and biomass burning-smoke exposition. We demonstrated the highly conserved composition of the Mexican Amerindian population. Although we found differences in demographics and exposure, we corrected data by logistic regression. Nevertheless, our study is not exempt from limitations; first of all, we need more clinical data, such as the number of exacerbations or predominant phenotypes (bronchitis or emphysema). We also require other auxiliary tools, for instance, expression-related or immunohistochemical. Additionally, we need to include more COPD patients to strengthen the severity analysis.

## 5. Conclusions

Single-nucleotide variants in *CYP2C8*, *CYP2C9*, and *MGST3* genes are associated with the risk of COPD secondary to biomass-burning smoke exposure. In addition, shared haplotype blocks in *MGST3* and *ARNT2* were found in both tobacco smokers and biomass-burning smoke-exposed subjects.

**Supplementary Materials:** Supplementary materials can be found at https://www.mdpi.com/article/10.3390/cimb45020053/s1. Supplementary Figure S1. Sample selection criteria for both comparison groups, Supplementary Figure S2. Sample selection flowchart for both comparison groups, Supplementary Figure S3. SNPs filtering flowchart for SNPs association analysis, Supplementary Figure S4. Multiple Correspondence Analysis (MCA), Supplementary Figure S5. Multiple correspondence analysis (MCA) for smokers' comparison, Supplementary Figure S6. Multiple correspondence analysis (MCA) for biomass' comparison, Supplementary Figure S7. Gene schemes with the SNPs associated in haplotype analysis in smokers' comparison, Supplementary Figure S8. Gene schemes with the SNPs associated with haplotype analysis in biomass comparison, Supplementary Table S1. List of SNPs included in the analysis, Supplementary Table S2. Molecular data for SNPs associated in the smokers' comparison, Supplementary Table S3. Molecular data for SNPs associated in exposed biomass-burning smoke comparison, Supplementary Table S4. Alleles data for smokers' comparison with no correction, Supplementary Table S5. Genotype data for smokers' comparison with no correction, Supplementary Table S6. Alleles data in exposed biomass-burning smoke comparison with no correction, Supplementary Table S7. Genotype data in exposed biomass-burning smoke comparison with no correction, Supplementary Table S8. Analysis by severity in patients with COPD-S, Supplementary Table S9. Analysis by severity in patients with COPD-BBES.

**Author Contributions:** Conceptualization, G.P.-R., A.R.-V., R.S., J.P.-R. and R.F.-V.; Data curation, G.P.-R., J.C.F.-L. and R.S.; Formal analysis, E.A.-O., G.P.-R., A.R.-V. and J.C.F.-L.; Funding acquisition, R.S. and R.F.-V.; Investigation, E.A.-O., A.R.-V., F.C.-V. and J.P.-R.; Methodology, E.A.-O., M.E.R.-D., M.d.L.M.-G. and R.F.-V.; Project administration, R.d.J.H.-Z., R.S., J.P.-R. and R.F.-V.; Resources, A.R.-V., R.d.J.H.-Z., M.E.R.-D., F.C.-V., M.d.L.M.-G. and R.F.-V.; Software, E.A.-O., G.P.-R., J.C.F.-L. and R.S.; Supervision, A.R.-V., M.d.L.M.-G. and R.F.-V.; Validation, E.A.-O., G.P.-R., R.d.J.H.-Z., R.S., J.P.-R. and R.F.-V.; Visualization, E.A.-O. and J.C.F.-L.; Writing—original draft, E.A.-O., G.P.-R., J.P.-R. and R.F.-V.; Writing—review and editing, E.A.-O. and R.F.-V. All authors have read and agreed to the published version of the manuscript.

**Funding:** This work was supported by the budget allocated to research (RFV-HLA Laboratory) from the Instituto Nacional de Enfermedades Respiratorias Ismael Cosío Villegas (INER).

**Institutional Review Board Statement:** The study was conducted according to the guidelines of the Declaration of Helsinki and approved by the Institutional Ethics Committee of Instituto Nacional de Enfermedades Respiratorias Ismael Cosío Villegas (protocol numbers C35-19).

**Informed Consent Statement:** Informed consent was obtained from all subjects involved in the study.

**Data Availability Statement:** Data analyzed in this study are available in ClinVar under the submission code SUB11745124 (SCV002540796–SCV002540803) and SUB11745177 (SCV002540844–SCV002540866).

**Acknowledgments:** The authors acknowledge the support received from physicians and technicians from the COPD clinic at INER to confirm the diagnosis, the acquisition of data on lung function, and the clinical care of the study participants.

**Conflicts of Interest:** The authors declare no conflict of interest.

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
