# Peer review of "SNPs Sets in Codifying Genes for Xenobiotics-Processing Enzymes Are Associated with COPD Secondary to Biomass-Burning Smoke"

_cimb, doi:10.3390/cimb45020053_

Round 1

Reviewer 1 Report

Major comments.

1.     While some procedures for quality control of genotype data for for genome-wide genotyping data are mentiones, quality control is not extensively described (i.e. assessment of sex discordance, call rate for SNPs,…). See PMID: 20718045, PMID: 29484742 for quality control procedures and STREGA guidelines for appropriate reporting (i.e. p-value threshold for HWE is not reported)

2.     The authors say that they used Fisher’s exact test (line 136) for association analysis but Tables exhibits logistic regression results. Were both methods explored? Why was an old version of PLINK used? PLINK v2 has been out for a while.

3.     Mexicans are admixed individuals, therefore, they have different proportions of African, Native American and European ancestry that could impact the results observed. Please, conduct ADMIXTURE analysis (PMID: 21682921) or principal component analysis (PCA) of the genome-wide genotyping data to assess genetic ancestry. Additionally, include PCs as covariates in your regression models to account for potential differences in allele frequencies due to differences in genetic ancestry that may impact the association results.

4.     Bonferroni correction may result in overcorrection since SNPs in the region may be in linkage disequilibrium. Alternatively, you could consider PMID: 16077740 to reduce the burden of multiple testing.

5.     Is there any in silico evidence of the functional role of these SNPs (see i.e., Phenoscanner, GTEx, Haploreg, Open Target Genetics, REALGAR) that support their implications in the traits?

6.     Comment on the study strengths/limitations in the discussion.

Minor comments.

-State what guidelines were used to define COPD.

-Further English editing is required.

-Were allele frequencies for Mexican consistent with those reported in Mexicans from 1000 Genomes project? Are there differences in allele frequencies in ancestral populations (Native American, European and African populations) that may indicate that these variants exhibit a gradient of effects in other populations?

Author Response

  1. While some procedures for quality control of genotype data for genome-wide genotyping data are mentioned, quality control is not extensively described (i.e. assessment of sex discordance, call rate for SNPs). See PMID: 20718045, PMID: 29484742 for quality control procedures and STREGA guidelines for appropriate reporting (i.e. p-value threshold for HWE is not reported)

We appreciate your valuable recommendations and bibliographic information. We rewrote and included all the QC considerations in the study (lines 128-131):

PLINK v. 1.07 [21] was used for data quality control (QC). We considered a genotype call rate >95% and eliminated subjects with > 0.05 of missing genotypes. Sex discrepancies were considered by X chromosome homozygosity (men > 0.8, women < 0.2), while relatedness was assessed by identity by descent (IBD) considering pi-hat values < 0.25.

For Hardy-Weinberg equilibrium, we added the p-values threshold in line 124.

  1. The authors say that they used Fisher’s exact test (line 136) for association analysis but Tables exhibits logistic regression results. Were both methods explored? Why was an old version of PLINK used? PLINK v2 has been out for a while.

Thanks for your question. We explored both methods, but we presented logistic regression results because it is a better association method since it includes the covariates in the analysis.

When we got the genotyping data, Plink 1.9 was a beta software. Results between versions do not differ, differences between versions are the improvement in the speed of data processing and some extra functions, or even some functions are no longer available (haplotypes function mainly).

  1. Mexicans are admixed individuals, therefore, they have different proportions of African, Native American and European ancestry that could impact the results observed. Please, conduct ADMIXTURE analysis (PMID: 21682921) or principal component analysis (PCA) of the genome-wide genotyping data to assess genetic ancestry. Additionally, include PCs as covariates in your regression models to account for potential differences in allele frequencies due to differences in genetic ancestry that may impact the association results.

We appreciate your valuable comments. We conducted PCA for ancestry composition and added as supplementary images. We did not include PCA in logistic regression due to overcorrection; the comparison group had similar ancestry composition. (lines 137-141)

Figure. PCA for the ancestral composition of the included population. We included Hapmap population data: northern Europeans from Utah (CEU), Han Chinese from Beijing (HCB), Japanese from Tokyo (JPT), and Yoruba in Ibadan from Nigeria (YRI). COPD_BBS: COPD patient exposed to biomass burning-smoke, COPD_S: COPD patient smokers, SWOC: Smoker without COPD, COPD_BBS: COPD secondary to biomass burning-smoke exposition, BBES: Biomass burning exposed subject.

Figure. Admixture plot for ancestry composition in subjects included. We included Hapmap population data: Northern Europeans from Utah (CEU), Han Chinese from Beijing (HCB), Japanese from Tokyo (JPT), and Yoruba in Ibadan from Nigeria (YRI). The plot was generated using 32 SNPs reported by Huerta-Chagoya and Cols. [A panel of 32 AIMs suitable for population stratification correction and global ancestry estimation in Mexican mestizos]. The biomass comparison group (COPD_BBS, BBES) was highly composed of Mexican Amerindian, while smokers´ comparison (COPD_S, SWOC) presented heterogenic composition, predominantly Caucasian and Mexican Amerindian information.

  1. Bonferroni correction may result in overcorrection since SNPs in the region may be in linkage disequilibrium. Alternatively, you could consider PMID: 16077740 to reduce the burden of multiple testing.

Thanks for the comment; we considered the overcorrection bias because we added 2 p-values for allele association: p-values adjusted by covariates and p-values corrected by the Bonferroni test.

  1. Is there any in silico evidence of the functional role of these SNPs (see i.e., Phenoscanner, GTEx, Haploreg, Open Target Genetics, REALGAR) that support their implications in the traits?

Thank you for the recommendations. We added a brief investigation of the principal association found in our study. (lines 446-449)

  1. Comment on the study strengths/limitations in the discussion.

Thanks for your recommendation. We added a paragraph with strengths/limitations (lines 460-467)

Reviewer 2 Report

Ambrocio-Ortiz and colleagues performed genome-wide exome genotyping of 1500 Mexican subjects. The authors investigated single nuclear polymorphisms (SNPs) associated with COPD due to cigarette smoking and biomass-burning smoke. The article is exciting and identifies several potential novel molecular targets. However, there are a few minor aspects that have to be addressed before the article can be considered for publication in Current Issues in Molecular Biology.

Comments

1.      The authors should emphasise the novel aspects of the conducted study compared to the SNPs analyses performed and published before. The authors should also briefly discuss the differences and similarities with the previous studies that focused on identifying SNPs associated with COPD.

2.      The authors should discuss the targets that were identified to associate with the disease severity.

3.      COPD is an umbrella term comprising airway disease, emphysema and alterations in the pulmonary vasculature. Have the authors looked into SNPs that are associated with, for example, predominant emphysema phenotype over bronchitis? 

Author Response

  1. The authors should emphasise the novel aspects of the conducted study compared to the SNPs analyses performed and published before. The authors should also briefly discuss the differences and similarities with the previous studies that focused on identifying SNPs associated with COPD.

Thank you for your recommendation. Now we added a brief paragraph talking about our study's main aspects. (lines 460-467)

  1. The authors should discuss the targets that were identified to associate with the disease severity.

We appreciate your valuable feedback. We know the importance of the main finding in the context of previous studies; we found moderate info about the severity and our SNPs and added it in the discussion section in the lines. (lines 252-258 results, lines 405-412 discussion)

  1. COPD is an umbrella term comprising airway disease, emphysema and alterations in the pulmonary vasculature. Have the authors looked into SNPs that are associated with, for example, predominant emphysema phenotype over bronchitis?

We do not have much information about clinical classification, so we did not include analysis with other clinical characteristics.

Author Response

  1. The title: single nucleotide polymorphisms are not found in enzymes, but in genes encoding them. Please rephrase.

Thank you for your recommendation. We rewrote the title

  1. Introduction: Emphysema is not present among the main features of COPD. In fact, it is stated that bronchitis is the predominant phenotype in COPD-BBS, but it is not further explained how that contrasts with the situation in smokers. Please explain this difference more in detail. Also, as you have significant differences between gender ratio in COPD-S and COPD-BBS groups, you should probably also state that phenotype in COPD might be influenced by gender: https://doi.org/10.1164/rccm.201512- 2379ED.

Thank you for your recommendation. Unfortunately, we do not have enough clinical info to analyze phenotypes extensively.

  1. Male/female ratio reported in table 1 for COPD-BBS and BBES is inconsistent to what you describe in the discussion, line 303. Please correct the mistake. If Table 1 shows correct information, then please also discuss this unusual finding.

Thanks for the observation. We rewrite the idea to clarify our statement

  1. Discussion, line 392: MGST1 is described as relevant to extracellular matrix reorganization, without the reference to support this. However, MGST1 is generally described as a protein localized to the endoplasmic reticulum and outer mitochondrial membrane, which is thought to protect these membranes from oxidative stress. It catalyzes the conjugation of glutathione to electrophiles and the reduction of lipid hydroperoxides. Also, MGST1 was found to inhibit ferroptosis, which could be of significance for COPD. Recently, it was reported that MGST1 is differentially expressed in AT2 cells from healthy and COPD subjects: https://doi.org/10.1038/s41467-022- 28062-9. Please extend this paragraph and also provide a reference for your description.

Thanks for your recommendation. We added a paragraph mentioning this information (lines 394-396)

  1. Would it make sense to perform analyses for COPD-S separated by sex? Some of the enzymes you investigated are differentially regulated between men and women, resulting in several-fold higher activity in women. Therefore, some genetic variants leading to impairment of enzyme activity might influence men more strongly.

Thanks for your recommendation, but we did not consider it necessary. Comparison by sex should be necessary for studies of mRNA quantification when sex could directly affect the levels. Even though, to avoid bias related to sex, we included sex in covariate to include possible effects in sex.

Round 2

Reviewer 1 Report

The authors have addressed my comment, and therefore, improved their manuscript as a result. One minor formatting and one minor methodological issue have been asked to be addressed further.

1. It is quite difficult to discern the different groups per color status, particularly COPD and HBC groups because of the color similiraties. Consider using a different color/point scheme.

2. In the admixture plot (Figure 1B) represent only the BBES, COPD and SWOC group since the ancestral groups are considered the reference.  How was K selected? For Mexican populations, K=3 is usually the best model, with inclusion of CEU, YRI and Native American data (NAM) (japanese data could alternatively be used as a proxy of NAM when NAM data is not available)

Author Response

  1. It is quite difficult to discern the different groups per color status, particularly COPD and HBC groups because of the color similiraties. Consider using a different color/point scheme.

We sincerely appreciate your recommendation. We changed the points’ colors and shapes and added an explanation.

  1. In the admixture plot (Figure 1B) represent only the BBES, COPD and SWOC group since the ancestral groups are considered the reference.  How was K selected? For Mexican populations, K=3 is usually the best model, with inclusion of CEU, YRI and Native American data (NAM) (japanese data could alternatively be used as a proxy of NAM when NAM data is not available)

Thank you for your accurate comments. We included the k value used (line 142). We did not mention this before, but we used k = 3, based on the descriptions by Huerta-Chagoya and Silva-Zolezzi and cols.